# Remimazolam-Based Anesthesia and Systemic Inflammatory Biomarkers in Relation to Postoperative Delirium in Elderly Patients: A Retrospective Cohort Study

**DOI:** 10.3390/medicina61061023

**Published:** 2025-05-30

**Authors:** Hayoung Lee, Keunyoung Kim, Cheol Lee

**Affiliations:** 1Department of Anesthesiology and Pain Medicine, Wonkwang University School of Medicine, 895 Muwang-ro, Iksan-si 54538, Jeonbuk, Republic of Korea; julia7071@naver.com; 2Department of General Surgery, Division of Coloproctology, Wonkwang University School of Medicine, 895 Muwang-ro, Iksan-si 54538, Jeonbuk, Republic of Korea; kky@wku.ac.kr

**Keywords:** remimazolam, postoperative delirium, neutrophil-to-lymphocyte ratio, systemic inflammation, elderly patients

## Abstract

*Background and Objectives*: Postoperative delirium (POD) is a common complication in elderly patients undergoing major surgery, associated with systemic inflammation and potentially influenced by anesthetic techniques. This study investigated whether remimazolam-based total intravenous anesthesia (R-TIVA) reduces the level of systemic inflammatory biomarkers and the incidence of POD more compared to inhalational anesthesia (IA) or balanced anesthesia (BA) in patients aged ≥ 65 years undergoing major non-neurosurgical, non-cardiac surgery. *Materials and Methods*: This retrospective cohort study analyzed the medical records of 340 patients categorized by anesthesia type: R-TIVA (*n* = 111), IA (*n* = 117), or BA (*n* = 112). Propensity score matching (PSM) created POD (*n* = 104) and No POD (*n* = 106) cohorts. Systemic inflammatory biomarkers—the systemic immune–inflammation index (SII), the neutrophil-to-lymphocyte ratio (NLR), the platelet-to-lymphocyte ratio (PLR), and C-reactive protein (CRP)—were measured pre- and postoperatively. POD was identified by clinical symptoms or the postoperative use of antipsychotics/sedatives. *Results*: The incidence of POD did not significantly differ among the R-TIVA, IA, and BA groups. However, the postoperative SII, NLR, PLR, and CRP levels were significantly lower in the R-TIVA group compared to those in the IA group (*p* < 0.05). Both the preoperative (*r_pb_* = 0.72, *p* < 0.01) and postoperative (*r_pb_* = 0.58, *p* < 0.01) NLRs were strongly correlated with POD. Higher NLR values predicted the incidence of POD, with odds ratios of 1.71 for preoperative and 1.32 for postoperative measurements. *Conclusions*: While R-TIVA did not significantly reduce the incidence of POD compared to that of IA or BA, it was associated with reduced levels of postoperative inflammatory biomarkers. The preoperative and postoperative NLRs emerged as strong predictors of POD, suggesting their potential utility in guiding prophylactic strategies for older surgical patients. These findings underscore the interplay between anesthesia type, systemic inflammation, and delirium risk.

## 1. Introduction

Postoperative delirium (POD) is a common complication in elderly patients undergoing major surgery, characterized by acute changes in cognition and attention, and is associated with increased morbidity, prolonged hospitalization, and cognitive decline [1,2]. The risk factors include advanced age, comorbidities, and systemic inflammation, defined by elevated biomarkers, such as the neutrophil-to-lymphocyte ratio (NLR) and C-reactive protein (CRP), which are associated with an increased postoperative delirium (POD) risk [3,4,5]. The anesthetic techniques may influence the incidence of postoperative delirium (POD), yet the studies comparing total intravenous anesthesia (TIVA), inhalational anesthesia (IA), and balanced anesthesia (BA) yield inconsistent findings [6,7,8,9].

Remimazolam, an ultra-short-acting benzodiazepine, may offer advantages over traditional benzodiazepines, which are linked to an increased delirium risk [10,11]. Its anti-inflammatory properties, including the suppression of tumor necrosis factor-alpha (TNF-α) and interleukin-6 (IL-6), could mitigate systemic inflammation and reduce the POD risk [12,13,14]. However, the impact of remimazolam-based TIVA (R-TIVA) on systemic inflammatory biomarkers and the incidence of POD in elderly patients remains unexplored. Balanced anesthesia (BA), combining remimazolam with volatile anesthetics (e.g., sevoflurane and desflurane), may reduce the anti-inflammatory benefits of remimazolam, as volatile anesthetics have pro-inflammatory effects that could counteract cytokine suppression (e.g., TNF-α and IL-6) [12,13,14]. This hypothesis warrants further investigation.

This study hypothesized that R-TIVA reduces the incidence of POD by lowering the levels of systemic inflammatory biomarkers—the systemic immune–inflammation index (SII), the NLR, the platelet-to-lymphocyte ratio (PLR), and C-reactive protein (CRP)—more compared to IA or BA in patients aged ≥ 65 years undergoing major non-cardiac, non-neurosurgical surgery. Using propensity score matching (PSM), we assessed the perioperative biomarker differences between the POD and No POD groups and evaluated their predictive value for POD.

## 2. Materials and Methods

### 2.1. Study Design and Patient Selection

This retrospective cohort study analyzed the medical records of patients aged ≥ 65 years who underwent major non-cardiac, non-neurosurgical procedures under general anesthesia at Wonkwang University Hospital from March 2022 to February 2024. This study was approved by the Institutional Review Board (IRB No. 2023-10-035, 10 October 2023) and conducted in accordance with the Declaration of Helsinki. Informed consent was waived due to the retrospective design. Due to the retrospective design, formal sample size calculation was not conducted. All eligible patients meeting the inclusion criteria from March 2022 to February 2024 were analyzed to maximize statistical power. Patients were included if they underwent major non-cardiac, non-neurosurgical procedures under general anesthesia and had complete perioperative data. The presence or absence of POD was later assessed as an outcome based on postoperative clinical records, not as an inclusion criterion.

### 2.2. Data Collection

The data included demographic characteristics (age, sex, body mass index [BMI], and American Society of Anesthesiologists [ASA] classification), perioperative details (surgery type, fluid administration, anesthesia/surgery duration, transfusion, hypotension, bradycardia, and vasoactive agent use), and comorbidities (diabetes mellitus, hypertension, coronary artery disease, chronic obstructive pulmonary disease, peripheral arterial occlusive disease, cerebrovascular accident, psychiatric disorder, and polypharmacy). Emergency surgery, hospitalization duration, cumulative patient-controlled analgesia (PCA) volume, and pre- and postoperative (day 1) biomarkers (SII, NLR, PLR, and CRP) were recorded. The data on Enhanced Recovery After Surgery (ERAS) protocols and intraoperative medications (e.g., dexmedetomidine, ketamine, and midazolam) were not consistently recorded due to the retrospective design and were excluded via analysis, representing a study limitation.

The patients were categorized into three groups: (1) R-TIVA (remimazolam-based), (2) IA (sevoflurane or desflurane), and (3) BA (remimazolam plus inhalational agents). The remimazolam dosages in the R-TIVA and BA groups were not standardized due to the retrospective design, with infusion rates ranging from 0.5 to 2 mg/kg/h, which may influence the outcomes. POD was defined by clinical symptoms (e.g., confusion, disorientation, agitation, and hallucinations) or the use of postoperative antipsychotics/sedatives, such as quetiapine, haloperidol, and midazolam. Diagnosis relied on medical record documentation, without standardized tools like the Confusion Assessment Method for Intensive Care Unit (CAM-ICU) [15] Biomarkers were calculated as follows: SII = (neutrophil count × platelet count) ÷ lymphocyte count; NLR = neutrophil count ÷ lymphocyte count; PLR = platelet count ÷ lymphocyte count.

### 2.3. Outcomes

The primary outcome was the association between systemic inflammatory biomarkers and the incidence of POD among the anesthetic techniques. The secondary outcomes included the incidence of POD, the predictive value of biomarkers for POD, and the perioperative biomarker differences between the POD and No POD groups after PSM.

### 2.4. Statistical Analysis

Analyses were performed using SPSS version 29 (IBM Corp., Armonk, NY, USA), with significance set at *p* < 0.05. Continuous variables are expressed as mean ± standard deviation (SD) or numbers (percentages) and were analyzed using one-way ANOVA (three-group comparisons) or independent *t*-tests (two-group comparisons). Categorical variables were assessed using chi-square or Mantel–Haenszel tests.

Point-biserial correlation (*r_pb_*) evaluated the biomarker–POD relationships, with strength interpreted per Cohen’s guidelines: *r_pb_* < 0.3 (weak), 0.3–0.5 (moderate), and >0.5 (strong) [16]. PSM used the 1:1 nearest neighbor method (caliper width 0.15), targeting standardized mean differences (SMD) < 0.1. Pre- and post-PSM SMDs were calculated for the covariates (age, sex, BMI, ASA, and surgery duration). Missing data were handled by listwise deletion, with sensitivity analyses confirming minimal bias. Logistic regression was adjusted for age, sex, BMI, ASA classification, and surgery duration. Receiver Operating Characteristic (ROC) curves determined biomarker the cut-offs, with sensitivity, specificity, and area under the curve (AUC) reported.

## 3. Results

### 3.1. Demographic and Perioperative Characteristics

Of the 340 patients, 111 received remimazolam-based total intravenous anesthesia (R-TIVA; *n* = 111), 117 received inhalational anesthesia (IA; *n* = 117), and 112 received balanced anesthesia (BA; *n* = 112). No significant differences were observed in age, sex, BMI, ASA classification, surgery type, fluid volume, anesthesia/surgery duration, transfusions, and comorbidities (*p* > 0.05; Table 1). The IA group had higher incidences of hypotension (*p* = 0.02), bradycardia (*p* = 0.02), and vasoactive agent use (*p* = 0.01) compared to those of R-TIVA. The preoperative biomarkers were similar (*p* > 0.05), but the postoperative SII, NLR, PLR, and CRP were significantly higher in the IA group versus R-TIVA (*p* < 0.05; Cohen’s d = 0.42–0.67).

### 3.2. Correlation Between Biomarkers and POD

The preoperative SII (*r_pb_* = 0.39, *p* < 0.01) and the postoperative SII (*r_pb_* = 0.36, *p* < 0.01) showed weak positive correlations with postoperative delirium (POD). The preoperative NLR (*r_pb_* = 0.72, *p* < 0.01) and the postoperative NLR (*r_pb_* = 0.58, *p* < 0.01) exhibited strong and moderate correlations, respectively. The preoperative and postoperative PLRs (*r_pb_* = 0.44 and 0.37, *p* < 0.01) and CRP (*r_pb_* = 0.42 and 0.34, *p* < 0.01) showed weak correlations (Table 2).

### 3.3. Predictive Value of Biomarkers for POD

Logistic regression identified the preoperative NLR (OR = 1.71, 95% CI: 1.46–2.00, *p* < 0.01) and the postoperative NLR (OR = 1.32, 95% CI: 1.13–1.55, *p* < 0.01) as strong predictors of postoperative delirium (POD). The postoperative CRP (OR = 1.13, 95% CI: 1.03–1.24, *p* = 0.01) was also predictive, while the SII and the PLR had limited utility due to the confidence intervals, including one (Table 3).

### 3.4. Propensity Score Matching Analysis

After PSM, variables such as surgery type, the fluid amount administered, diabetes mellitus, hypertension, chronic obstructive pulmonary disease, peripheral arterial occlusive disease, and cumulative PCA volume showed no significant differences between the POD and No POD groups (*p* > 0.05). However, sex (*p* = 0.04) and BMI (*p* = 0.01) remained significantly different post-matching, with males and those with a higher BMI being more prevalent in the POD group. This imbalance may reflect residual confounding factors and are addressed in the limitations. Significant differences persisted in age, ASA classification, transfusion, the duration of anesthesia, and surgery before and after matching (*p* < 0.01). Before matching, the incidences of coronary artery disease differed significantly between the groups (*p* < 0.01), but not after (*p* = 0.12). The values for psychiatric disease, cerebrovascular accidents, polypharmacy, emergency surgery, hospitalization, the preoperative and postoperative SIIs, the NLR, the PLR, and the postoperative CRP were significantly higher in the POD group (*p* < 0.01) before and after matching. The preoperative CRP level differed significantly post-matching (*p* < 0.01), but not before (*p* = 0.12) (Table 4).

### 3.5. Predictive Cut-Offs

ROC analysis showed the preoperative NLR (AUC = 0.93, 95% CI: 0.90–0.96) and the postoperative NLR (AUC = 0.86, 95% CI: 0.82–0.90) as strong predictors of POD. The cut-offs were ≥4.2 (sensitivity: 85%; specificity: 90%) for the preoperative NLR and ≥6.9 (sensitivity: 80%; specificity: 85%) for the postoperative NLR (Figure 1).

## 4. Discussion

This study demonstrated that patients receiving remimazolam-based total intravenous anesthesia (R-TIVA) had significantly lower levels of systemic inflammatory biomarkers, including the SII, the NLR, the PLR, and CRP, compared to those receiving inhalational anesthesia (IA). Despite this reduction, there was no significant difference in the incidence of postoperative delirium (POD) among the R-TIVA (30.6%), IA (33.3%), and balanced anesthesia (BA) (29.5%) groups (*p* = 0.82). The lack of difference in the incidence of POD suggests that systemic inflammation is only one of several factors contributing to delirium. The anti-inflammatory properties of remimazolam, which suppress mediators like TNF-α and IL-6, likely account for the observed biomarker reductions [12,13,14]. These findings highlight the complex interplay between anesthesia type, inflammation, and POD risk, indicating that R-TIVA’s anti-inflammatory effects do not directly translate to reduced delirium incidence. The lack of reduced POD incidence despite lower systemic inflammation in the R-TIVA group suggests that the systemic biomarkers (e.g., NLR and CRP) may not fully reflect neuroinflammation, which is likely a key driver of POD pathogenesis, highlighting the multifactorial nature of delirium [17,18].

These findings are clinically important because they suggest that monitoring systemic inflammatory biomarkers, particularly the NLR, could identify elderly surgical patients at increased risk of developing POD. The preoperative and postoperative NLRs showed strong correlations with POD (*r_pb_* = 0.72, *p* < 0.01, and *r_pb_* = 0.58, *p* < 0.01), with AUCs of 0.93 and 0.86, respectively, indicating high predictive power. These biomarkers could guide targeted interventions, such as preoperative anti-inflammatory therapies (e.g., corticosteroids, non-steroidal anti-inflammatory drugs [NSAIDs]) and enhanced postoperative monitoring (e.g., [CAM-ICU] and neurocognitive assessments), to mitigate the POD risk [5]. By identifying high-risk patients early, clinicians can implement strategies to improve outcomes, reduce hospital stays, and lower the healthcare costs associated with POD. However, the multifactorial nature of POD, including patient-specific factors like age and comorbidities, underscores the need for a comprehensive approach beyond biomarker monitoring [17,18].

Another important finding is the significant association between POD and patient-specific and operation-specific risk factors, including older age, male sex, a higher BMI, a worse ASA classification, longer anesthesia/surgery duration, transfusions, and comorbidities like psychiatric disorders and cerebrovascular accidents. These factors were more prevalent in the POD group both before and after PSM, suggesting a significant link between heightened inflammation and delirium development. The inflammatory biomarkers (SII, NLR, PLR, and CRP) were consistently elevated in the POD group, reinforcing inflammation’s role in delirium pathogenesis. The higher incidence of hemodynamic events (e.g., hypotension and bradycardia) in the IA group may have contributed to the elevated biomarkers, reflecting increased surgical stress [19]. These findings emphasize the need to address modifiable risk factors, such as optimizing surgical duration and transfusion protocols, to reduce the POD risk [20,21].

Compared to the prior studies, our findings align with Song et al. (2022), who reported that the systemic immune–inflammation index (SII) predicted POD in elderly surgical patients, but found no direct link between reduced inflammation and a lower incidence of POD [17]. Similarly, He et al. (2020) identified the preoperative NLR as a significant predictor of POD in elderly patients with a hip fracture (AUC = 0.79), though our study reports higher AUCs (0.93 and 0.86), possibly due to the differences in the patient populations or biomarker timings [22]. Unlike these studies, our focus on remimazolam-based TIVA versus IA and BA highlights the unique anti-inflammatory profile of remimazolam, which was not observed with BA, likely due to the variable remimazolam dosages. The differences in the AUC values and the NLR thresholds may reflect geographic or ethnic variability, statistical methods, or advances in understanding NLR’s role in inflammation [23,24,25,26,27]. These comparisons underscore the need for further validation among diverse settings to confirm the generalizability of our findings.

This study has several limitations. Its retrospective design relies on medical record accuracy, risking selection bias from non-random anesthetic assignment and information bias from incomplete POD documentation, with 10% of the patients excluded due to missing biomarker data. The single-center setting in South Korea may limit generalizability due to variations in anesthesia protocols or population-specific factors. POD diagnosis relied on documented symptoms (e.g., confusion, disorientation, agitation, and hallucinations) or postoperative antipsychotic/sedative use, as the CAM-ICU could not be used, increasing the misclassification risk since the symptoms may be non-specific and medications may be used for non-delirium indications [15]. The postoperative biomarkers were measured on day 1, potentially missing the peak inflammatory responses (24–72 h), and the remimazolam dosages in the R-TIVA and BA groups were not standardized due to the retrospective design, with the infusion rates ranging from 0.5 to 2 mg/kg/h, which may influence the observed anti-inflammatory effects and POD outcomes. The mean infusion rate was approximately 1.2 ± 0.4 mg/kg/h based on the available data, although full consistency across all the patients could not be verified retrospectively.

Unmeasured confounders, including postoperative pain, infection, ERAS protocol application, and intraoperative medications (e.g., dexmedetomidine and ketamine), may have influenced the biomarker levels and POD incidence. The lack of standardized delirium assessments, such as the CAM-ICU, introduces the risk of misclassification, as POD diagnosis relied on documented symptoms (e.g., confusion, disorientation, agitation, and hallucinations) or postoperative antipsychotic/sedative use. The symptoms may be non-specific, overlapping with emergence agitation, and medications may be used for non-delirium indications, increasing the diagnostic uncertainty [16].

Our PSM analysis matched patients based on POD status to evaluate biomarker differences, controlling for covariates like age, sex, BMI, ASA classification, and surgery duration. However, matching on anesthesia type (R-TIVA, IA, and BA) could have better isolated the effects of remimazolam on POD and biomarkers by reducing confounding from non-random treatment allocation. While this approach was not feasible in our retrospective design, we recommend it for future prospective studies to improve causal inference

## 5. Conclusions

Remimazolam-based TIVA was associated with reduced level of postoperative systemic inflammatory biomarkers (SII, NLR, PLR, and CRP) compared to those of IA, likely due to its GABA-A receptor agonism and reduced sympathetic activation, which may suppress pro-inflammatory cytokines (e.g., TNF-α and IL-6). However, this anti-inflammatory effect did not translate into a lower incidence of POD.

## Figures and Tables

**Figure 1 medicina-61-01023-f001:**
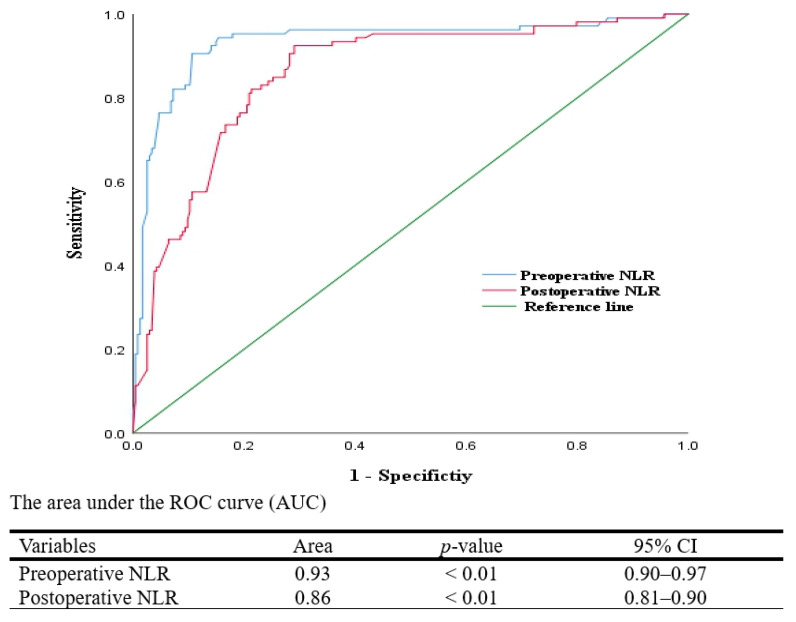
Receiver Operating Characteristic (ROC) curves for systemic inflammatory biomarkers predicting postoperative delirium (POD). AUC: area under curve; CI: confidence interval.

**Table 1 medicina-61-01023-t001:** Demographic and perioperative data.

	Group	Remimazolam-Based TIVA Group (*n* = 111)	Inhalational Anesthesia Group (*n* = 117)	Balanced Anesthesia Group (*n* = 112)	*p*-Value
Variables					
Age (years)		69.3 ± 3.0	70.1 ± 3.1	69.7 ± 3.5	>0.05
Sex (M/F)		58/53	60/57	59/53	>0.05
Body mass index (kg/m^2^)		23.6 ± 1.9	23.8 ± 1.8	23.7 ± 1.8	>0.05
ASA classification (I/II/III/IV)		12/44/49/6	12/44/53/8	9/46/49/8	>0.05
Type of surgery					>0.05
Urologic		33 (29.7)	38 (32.5)	30 (26.8	
Gastrointestinal		35 (31.5)	37 (31.6)	36 (32.1)	
Colorectal		34 (30.6)	33 (28.2)	33 (29.5)	
Gynecologic		9 (8.1)	9 (7.7)	13 (11.6)	
Fluid amount administered (mL)		1962.2 ± 197.3	1969.2 ± 189.6	1974.1 ± 184.4	>0.05
Duration of anesthesia (min)		203.9 ± 21.5	202.6 ± 19.7	205.7 ± 18.8	>0.05
Duration of surgery (min)		178.6 ± 20.8	177.2 ± 19.4	179.9 ± 18.6	>0.05
Transfusion (packed RBC > 3)		21 (18.9)	23 (19.7)	25 (22.3)	>0.05
HOT		12 (10.8)	29 (24.8)	18 (16.1)	0.02
Bradycardia		8 (7.2)	23 (19.7)	15 (13.4)	0.02
Vasoactive agents		10 (9.0)	26 (22.2)	14 (12.5)	0.01
DM		35 (31.5)	35 (29.9)	37 (33.0)	>0.05
HTN		45 (40.5)	46 (39.3)	44 (39.3	>0.05
CAD		5 (4.5)	5 (4.3)	4 (3.6)	>0.05
COPD		6 (5.4)	6 (5.1)	3 (2.7)	>0.05
PAOD		18 (16.2)	24 (20.5)	22 (19.6)	>0.05
Psychiatric disease		16 (14.4)	15 (12.8)	14 (12.5)	>0.05
CVA		22 (19.8)	29 (24.8)	24 (21.4)	>0.05
Polypharmacy		23 (20.7)	19 (16.2)	24 (21.4)	>0.05
Emergency surgery		12 (10.8)	16 (13.7)	12 (10.7)	>0.05
Hospitalization (days)		8.7 ± 1.7	9.0 ± 1.5	8.7 ± 1.4	>0.05
Postoperative delirium		34 (30.6)	39 (33.3)	33 (29.5)	>0.05
Cumulative PCA volume (mL)		343.3 ± 34.7	330.5 ± 27.2	329.7 ± 39.1	>0.05
Preoperative SII (×10^9^/L)		388.8 ± 147.5	389.4 ± 174.0	392.4 ± 179.4	>0.05
Postoperative SII (×10^9^/L)		562.9 ± 151.0	625.6 ± 133.4 *	603.0 ± 165.5	<0.01
Preoperative NLR		4.5 ± 4.1	4.9 ± 4.1	4.2 ± 3.2	>0.05
Postoperative NLR		6.0 ± 4.1	8.3 ± 6.2 *	6.9 ± 4.5	<0.01
Preoperative PLR		166.6 ± 74.1	172.0 ± 81.7	170.1 ± 89.2	>0.05
Postoperative PLR		204.5 ± 108.2	240.9 ± 125.5 *	217.9 ± 97.9	0.04
Preoperative CRP (mg/L)		3.9 ± 2.1	3.8 ± 2.4	3.9 ± 2.6	>0.05
Postoperative CRP (mg/L)		6.5 ± 3.1	7.7 ± 3.7 *	6.8 ± 3.5	0.03

Data are expressed as mean ± SD or numbers (%). TIVA, total intravenous anesthesia; ASA, American Society of Anesthesiologists; HOT, hypotension; DM, diabetes mellitus; HTN, hypertension; CAD, coronary arterial disease; COPD, chronic obstructive pulmonary disease; PAOD, peripheral arterial obstructive disease; CVA, cerebrovascular accident; PCA, patient-controlled analgesia; SII, systemic immune–inflammation index; NLR, neutrophil-to-lymphocyte ratio; PLR, platelet-to-lymphocyte ratio; CRP, C-reactive protein. * *p* vs. remimazolam-based TIVA group.

**Table 2 medicina-61-01023-t002:** Correlation between systemic inflammatory biomarkers and incidence of postoperative delirium.

	Pre Operative SII	Post Operative SII	Pre Operative NLR	Post Operative NLR	Pre Operative PLR	Post Operative PLR	Pre Operative CRP	Post Operative CRP
POD	*r_pb_* = 0.39	*r_pb_* = 0.36	*r_pb_* = 0.72	*r_pb_* = 0.58	*r_pb_* = 0.44	*r_pb_* = 0.37	*r_pb_* = 0.42	*r_pb_* = 0.34
	*p* < 0.01	*p* < 0.01	*p* < 0.01	*p* < 0.01	*p* < 0.01	*p* < 0.01	*p* < 0.01	*p* < 0.01

POD, postoperative delirium; SII, systemic immune–inflammation index; NLR, neutrophil-to-lymphocyte ratio; PLR, platelet-to-lymphocyte ratio; CRP, C-reactive protein.

**Table 3 medicina-61-01023-t003:** Logistic regression analysis of systemic inflammatory biomarkers associated with postoperative delirium.

	Univariate			Multivariate		
Systemic Inflammatory Biomarkers	OR	*p*-Value	95% CI	OR	*p*-Value	95% CI
Preoperative SII	1.01	<0.01	1.00–1.01	1.00	0.03	1.00–1.01
Postoperative SII	1.01	<0.01	1.00–1.01	1.00	0.01	1.00–1.01
Preoperative NLR	2.03	<0.01	1.75–2.37	1.71	<0.01	1.46–2.00
Postoperative NLR	1.35	<0.01	1.24–1.47	1.32	<0.01	1.13–1.55
Preoperative PLR	1.01	<0.01	1.00–1.01	1.01	0.05	1.00–1.01
Postoperative PLR	1.01	<0.01	1.00–1.01	1.00	0.23	0.99–1.00
Preoperative CRP	1.05	<0.01	0.97–1.13	0.92	0.43	0.74–1.13
Postoperative CRP	1.33	<0.01	1.25–1.42	1.13	0.01	1.03–1.24

OR, odds ratio; CI, confidence interval; SII, systemic immune–inflammation index; NLR, neutrophil-to-lymphocyte ratio; PLR, platelet-to-lymphocyte ratio; CRP, C-reactive protein.

**Table 4 medicina-61-01023-t004:** Demographic and perioperative data in postoperative delirium after propensity score matching.

	Before Matching			After Matching		
	No POD (*n* = 234)	POD *(n* = 106)	*p*-Value	No POD (*n* = 106)	POD *(n* = 106)	*p*- Value
Age (years)	69.1 ± 2.7	71.0 ± 3.8	<0.01	69.3 ± 2.9	71.0 ± 3.8	<0.01
Sex (M/F)	113/121	64/62	>0.05	56/50	64/42	0.04
Body mass index (kg/m^2^)	23.6 ± 1.8	23.9 ± 1.7	0.12	23.2 ± 1.8	23.9 ± 1.7	<0.01
ASA classification (I/II/III/IV)	28/117/81/8	5/17/70/14	<0.01	10/41/51/4	5/17/70/14	<0.01
Type of surgery			>0.05			>0.05
Urologic	66 (28.2)	35 (33.3)		34 (32.1)	35 (33.0)	
Gastrointestinal	81 (34.6)	27 (25.5)		36 (34.0)	27 (25.5)	
Colorectal	67 (28.6)	33 (31.1)		29 (27.3)	33 (31.1)	
Gynecologic	20 (8.5)	11 (10.4)		7 (6.6)	11 (10.4)	
Fluid amount administered (mL)	1971.8 ± 188.0	1961.3 ± 195.0	>0.05	1971.2 ± 188.3	1961.3 ± 195.0	>0.05
Duration of anesthesia (min)	195.1 ± 12.1	223.8 ± 19.8	<0.01	192.5 ± 11.6	223.8 ± 19.8	<0.01
Duration of surgery (min)	169.7 ± 11.9	198.1 ± 19.0	<0.01	167.4 ± 11.8	198.1 ± 19.0	<0.01
Transfusion (packed RBC > 3)	27 (11.5)	42 (39.6)	<0.01	10 (9.4)	42 (39.6)	<0.01
HOT	19 (7.8)	29 (26.1)	<0.01	15 (14.2)	29 (26.1)	0.02
Bradycardia	8 (3.3)	17 (15.3)	<0.01	6 (5.7)	17 (15.3)	0.01
Vasoactive agents	18 (7.3)	25 (22.5)	<0.01	12 (11.3)	25 (22.5)	0.02
DM	76 (32.5)	31 (29.2)	>0.05	39 (36.8)	31 (29.2)	>0.05
HTN	91 (38.9)	44 (41.5)	>0.05	45 (42.5)	44 (41.5)	>0.05
CAD	5 (2.1)	9 (8.5)	<0.01	4 (3.8)	9 (8.5)	>0.05
COPD	13 (5.6)	2 (1.9)	>0.05	9 (8.5)	2 (1.9)	>0.05
PAOD	43 (18.4)	21 (19.8)	>0.05	20 (18.9)	21 (19.8)	>0.05
Psychiatric disease	17 (7.3)	28 (26.4)	<0.01	8 (7.5)	28 (26.4)	<0.01
CVA	33 (14.1)	42 (39.6)	<0.01	15 (14.2)	42 (39.6)	<0.01
Polypharmacy	15 (6.4)	43 (40.6)	<0.01	7 (6.6)	43 (40.6)	<0.01
Emergency surgery	9 (3.4)	27 (25.5)	<0.01	4 (3.8)	27 (25.5)	<0.01
Hospitalization (days)	8.0 ± 0.6	11.0 ± 1.4	<0.01	7.9 ± 0.7	10.6 ± 1.4	<0.01
Cumulative PCA volume (ml)	334.4 ± 30.8	334.5 ± 31.5	>0.05	341.9 ± 34.0	334.5 ± 31.5	>0.05
Preoperative SII (×10^9^/L)	346.6 ± 132.5	486.6 ± 194.1	<0.01	384.3 ± 138.8	486.6 ± 194.1	<0.01
Postoperative SII (×10^9^/L)	561.4 ± 150.0	677.7 ± 124.0	<0.01	539.0 ± 147.9	677.7 ± 124.0	<0.01
Preoperative NLR	2.6 ± 1.9	8.6 ± 3.9	<0.01	2.7 ± 2.0	8.6 ± 3.9	<0.01
Postoperative NLR	5.1 ± 3.8	11.5 ± 5.0	<0.01	4.9 ± 3.3	11.5 ± 5.0	<0.01
Preoperative PLR	149.6 ± 70.6	213.8 ± 87.3	<0.01	150.9 ± 71.8	213.8 ± 87.3	<0.01
Postoperative PLR	203.1 ± 113.7	262.0 ± 97.3	<0.01	192.2 ± 126.4	262.0 ± 97.3	<0.01
Preoperative CRP (mg/L)	3.6 ± 4.8	4.4 ± 2.4	0.12	3.4 ± 1.9	4.4 ± 2.4	<0.01
Postoperative CRP (mg/L)	6.0 ± 3.1	9.2 ± 3.4	<0.01	5.9 ± 3.0	9.2 ± 3.4	<0.01

Data are expressed as mean ± SD or numbers (%). POD, postoperative delirium; ASA, American Society of Anesthesiologists; HOT, hypotension; DM, diabetes mellitus; HTN, hypertension; CAD, coronary arterial disease; COPD, chronic obstructive pulmonary disease; PAOD, peripheral arterial occlusive disease; CVA, cerebrovascular accident; PCA, patient-controlled analgesia; SII, systemic immune–inflammation index; NLR, neutrophil-to-lymphocyte ratio; PLR, platelet-to-lymphocyte ratio; CRP, C-reactive protein.

## Data Availability

The datasets analyzed are available from the corresponding author upon reasonable request.

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
