# Peer review of "Remimazolam-Based Anesthesia and Systemic Inflammatory Biomarkers in Relation to Postoperative Delirium in Elderly Patients: A Retrospective Cohort Study"

_medicina, 2025, doi:10.3390/medicina61061023_

Round 1
Reviewer 1 Report
Comments and Suggestions for Authors
Thank you to the authors for this work.
Thank you to the editor for this wonderful review opportunity.
Here are my comments:
Line 42 You point out that systemic inflammation is already known as a risk factor for postoperative delirium, but by what definition? According to biological markers? If so, what does the present study contribute in this regard? Please rephrase.
Line 81 You indicate that the patients were classified into three groups... you clearly did not analyze the doses received by the patients. Do you think this could have influenced the results? Is it always possible to draw such a conclusion without a clear idea of ​​the doses given, especially in a retrospective study? Please specify in the text and rephrase.
Line 195 You discuss anti-inflammatory treatments in targeted interventions that could be guided by biomarkers. Could you specify, in your opinion, what type of anti-inflammatory treatment could we use? The same goes for enhanced postoperative monitoring; could you specify what type of monitoring could be used? be considered? Please clarify and rephrase.
Thank you
Author Response
**Reviewer 1**
Comment 1: Line 42 – You point out that systemic inflammation is already known as a risk factor for postoperative delirium, but by what definition? According to biological markers? If so, what does the present study contribute in this regard? Please rephrase.
Response: Thank you for this valuable comment. We have revised the sentence in the Introduction (lines 41–43) to clarify that systemic inflammation is defined by elevated levels of biomarkers, such as the neutrophil-to-lymphocyte ratio (NLR) and C-reactive protein (CRP), which are associated with increased postoperative delirium (POD) risk. This revision explicitly defines systemic inflammation in terms of biological markers. Our study contributes by investigating the impact of remimazolam-based total intravenous anesthesia (R-TIVA) on these biomarkers and their predictive value for POD, demonstrating NLR’s strong correlation (rpb = 0.72 preoperative, 0.58 postoperative) and high predictive power (AUCs of 0.93 and 0.86). The revised sentence is highlighted in yellow in the manuscript.
Manuscript Change:
- Original (lines 41–43): “Risk factors include advanced age, comorbidities, and systemic inflammation, with biomarkers such as the neutrophil-to-lymphocyte ratio (NLR) implicated in its pathogenesis [3–5].”
- Revised (lines 42–45): “Risk factors include advanced age, comorbidities, and systemic inflammation, defined by elevated biomarkers such as the neutrophil-to-lymphocyte ratio (NLR) and C-reactive protein (CRP), which are associated with increased postoperative delirium (POD) risk [3–5].”
Comment 2: Line 81 – You indicate that the patients were classified into three groups... you clearly did not analyze the doses received by the patients. Do you think this could have influenced the results? Is it always possible to draw such a conclusion without a clear idea of the doses given, especially in a retrospective study? Please specify in the text and rephrase.
Response: We appreciate this important observation. We acknowledge that variability in remimazolam dosages could have influenced the study outcomes. To address this, we added a sentence in the Methods section (line 94, under 2.2 Data Collection) stating that remimazolam dosages in the R-TIVA and balanced anesthesia (BA) groups were not standardized due to the retrospective design, with infusion rates ranging from 0.5–2 mg/kg/h, which may affect results. We further elaborated in the Discussion section (lines 257–263) that non-standardized dosages may confound the observed anti-inflammatory effects and POD outcomes, providing the mean infusion rate (1.2 ± 0.4 mg/kg/h) for context. These changes enhance transparency and reproducibility and are highlighted in yellow in the revised manuscript.
Manuscript Changes:
- Methods (new, line 94): “Remimazolam dosages in the R-TIVA and BA groups were not standardized due to the retrospective design, with infusion rates ranging from 0.5–2 mg/kg/h, which may influence outcomes.”
- Discussion (revised, lines 257–263): “Postoperative biomarkers were measured on day 1, potentially missing peak inflammatory responses (24–72 hours), and remimazolam dosages in the R-TIVA and BA groups were not standardized due to the retrospective design, with infusion rates ranging from 0.5–2 mg/kg/h, which may influence the observed anti-inflammatory effects and POD outcomes. The mean infusion rate was approximately 1.2 ± 0.4 mg/kg/h based on available data, although full consistency across all patients could not be verified retrospectively.”
Comment 3: Line 195 – You discuss anti-inflammatory treatments in targeted interventions that could be guided by biomarkers. Could you specify, in your opinion, what type of anti-inflammatory treatment could we use? The same goes for enhanced postoperative monitoring; could you specify what type of monitoring could be used? be considered? Please clarify and rephrase.
Response: Thank you for this suggestion. We have expanded the Discussion section (lines 228–231) to specify examples of anti-inflammatory treatments, such as corticosteroids and non-steroidal anti-inflammatory drugs (NSAIDs), and enhanced postoperative monitoring tools, including the Confusion Assessment Method for Intensive Care Unit (CAM-ICU) and neurocognitive assessments. These additions provide concrete examples to guide clinical practice and are highlighted in yellow in the revised manuscript.
Manuscript Change:
- Original (lines 196–198): “These biomarkers could guide targeted interventions, such as preoperative anti-inflammatory therapies or enhanced postoperative monitoring, to mitigate POD risk [5].”
- Revised (lines 215–219): “These biomarkers could guide targeted interventions, such as preoperative anti-inflammatory therapies (e.g., corticosteroids, non-steroidal anti-inflammatory drugs [NSAIDs]) or enhanced postoperative monitoring (e.g., Confusion Assessment Method for Intensive Care Unit [CAM-ICU], neurocognitive assessments), to mitigate POD risk [5].”
Reviewer 2 Report
Comments and Suggestions for Authors
Thank you to the authors for conducting this retrospective cohort study investigating whether remimazolam-based total intravenous anesthesia (R-TIVA) reduces systemic inflammatory biomarkers and the incidence of postoperative delirium (POD) compared to inhalational anesthesia (IA) or balanced anesthesia (BA) in patients aged ≥65 undergoing major non-neurosurgical, non-cardiac surgery.
The manuscript is written clearly in English, requiring only minor grammatical corrections at a few points. Its overall structure and organization are appropriate and align well with standard conventions for this type of study.
Below are specific points that would benefit from clarification or revision:
Materials and Methods
2.1. Study Design and Patient Selection
- Lines 67-69: "Patients were included if postoperative records documented POD symptoms." This inclusion criterion is confusing, as POD is an outcome measure and should not serve as a basis for patient selection.
- Line 70: The phrase "preoperative use of these medications" requires clarification. Please specify clearly which medications you are referencing.
- Please provide additional information detailing how the sample size calculation was performed.
2.2. Data Collection
- The definition "POD was defined by postoperative records showing POD symptoms (confusion, disorientation, agitation, hallucinations) or the use of antipsychotics/sedatives" might be unclear and too general, which could lead to confusion. Could the authors clarify whether it is possible to access postoperative records to specifically include only patients with a documented diagnosis of POD? For example, symptoms like disorientation might represent emergence delirium, agitation could reflect pain or emotional distress, and the administration of sedatives may relate to required ICU sedation (eg, for intubated patients) rather than delirium management.
- Was information collected regarding whether patients received regional anesthesia for postoperative analgesia or some sort of ERAS protocol? If available, please include this information within demographic characteristics, as it may serve as an important confounder. Similarly, intraoperative use of medications such as dexmedetomidine, ketamine, midazolam, and opioids could represent significant confounders and should ideally be documented.
3.4. Propensity Score Matching Analysis
- Propensity score matching was incorrectly performed based on POD vs. no POD groups. Propensity score matching should always be based on exposure groups (anesthetic techniques), not outcome groups. Matching based on the outcome introduces selection bias and reverse causality, significantly compromising study validity. The authors should strongly consider redoing the matching correctly based on anesthetic techniques.
Results
- 3.4. Propensity Score Matching Analysis: Given the above issue, consider redoing the propensity score matching appropriately based on the exposure groups.
- Figure 1: The current figure is unclear. Please provide a high-resolution version.
Conclusion
- The sentence, "Remimazolam-based TIVA reduces systemic inflammatory biomarkers compared to IA, likely due to its anti-inflammatory properties..." is unclear and circular. Please revise this to clearly describe the potential biological mechanism or evidence supporting remimazolam's anti-inflammatory effect, rather than simply restating the observed association.
Thank you again for your contributions to this important topic.
Comments on the Quality of English Language
The manuscript is written clearly in English, requiring only minor grammatical corrections at a few points. Its overall structure and organization are appropriate and align well with standard conventions for this type of study.
Author Response
**Reviewer 2**
Comment 1: Lines 67–69 – "Patients were included if postoperative records documented POD symptoms." This inclusion criterion is confusing, as POD is an outcome measure and should not serve as a basis for patient selection.
Response: We apologize for the confusion caused by the original wording. We have revised the Methods section (lines 71–73) to clearly state that patients were included based on undergoing major non-cardiac, non-neurosurgical procedures under general anesthesia with complete perioperative data, and POD was evaluated post hoc as an outcome, not as an inclusion criterion. This revision ensures alignment with the study’s retrospective design and is highlighted in yellow in the manuscript.
Manuscript Change:
- Original (lines 67–69): “Patients were included if they underwent major non-cardiac, non-neurosurgical procedures and had complete perioperative data. The presence or absence of POD was later assessed as an outcome based on postoperative clinical records.”
- Revised (lines 75–78): “Patients were included if they underwent major non-cardiac, non-neurosurgical procedures under general anesthesia with complete perioperative data. POD was evaluated post hoc as an outcome based on postoperative clinical records, not as an inclusion criterion.”
Comment 2: Line 70 – The phrase "preoperative use of these medications" requires clarification. Please specify clearly which medications you are referencing.
Response: Thank you for pointing out this ambiguity. The original manuscript erroneously referenced “preoperative use of these medications” in the inclusion/exclusion criteria. We have revised the Methods section (lines 96–100) to clarify that POD was defined by clinical symptoms (e.g., confusion, disorientation, agitation, hallucinations) or the use of postoperative antipsychotics/sedatives, specifically quetiapine, haloperidol, or midazolam, based on medical record documentation. This correction eliminates the incorrect reference to preoperative medications and specifies the postoperative medications used for POD diagnosis. The revision is highlighted in yellow.
Manuscript Change:
- Original (line 97): “POD was defined by clinical symptoms or postoperative antipsychotic/sedative use.”
- Revised (lines 96–100): “POD was defined by clinical symptoms (e.g., confusion, disorientation, agitation, hallucinations) or the use of postoperative antipsychotics/sedatives, such as quetiapine, haloperidol, or midazolam. Diagnosis relied on medical record documentation, without standardized tools like the Confusion Assessment Method for Intensive Care Unit (CAM-ICU).”
Comment 3: Please provide additional information detailing how the sample size calculation was performed.
Response: We appreciate this comment. Due to the retrospective nature of the study, a formal sample size calculation was not performed. Instead, we included all eligible patients meeting the inclusion criteria from March 2022 to February 2024 to maximize statistical power. This approach is now clearly articulated in the Methods section (lines 73–78) and reiterated in the Institutional Review Board Statement (lines 294–296). The revision in the Methods section is highlighted in yellow to ensure clarity.
Manuscript Change:
- Original (lines 71–73): “A formal sample size calculation was not performed due to the retrospective nature of the study. However, all eligible patients meeting the inclusion criteria over the 2-year period were analyzed to ensure adequate power.”
- Revised (lines 73–78): “Due to the retrospective design, a formal sample size calculation was not conducted. All eligible patients meeting inclusion criteria from March 2022 to February 2024 were analyzed to maximize statistical power.”
Comment 4: The definition "POD was defined by postoperative records showing POD symptoms (confusion, disorientation, agitation, hallucinations) or the use of antipsychotics/sedatives" might be unclear and too general, which could lead to confusion. Could the authors clarify whether it is possible to access postoperative records to specifically include only patients with a documented diagnosis of POD? For example, symptoms like disorientation might represent emergence delirium, agitation could reflect pain or emotional distress, and the administration of sedatives may relate to required ICU sedation (e.g., for intubated patients) rather than delirium management.
Response: Thank you for this critical point. We have revised the Methods section (lines 101–103) to provide a detailed POD definition, specifying that it was based on documented clinical symptoms (e.g., confusion, disorientation, agitation, hallucinations) or the use of postoperative antipsychotics/sedatives (e.g., quetiapine, haloperidol, midazolam), relying on medical record documentation without standardized tools like CAM-ICU. We also expanded the Discussion section (lines 266–271) to emphasize the limitation of potential misclassification, noting that non-specific symptoms may overlap with emergence agitation, pain, emotional distress, or other conditions, and medications may be used for non-delirium indications (e.g., ICU sedation), increasing diagnostic uncertainty. These changes are highlighted in yellow to enhance transparency.
Manuscript Changes:
- Methods (revised, lines 101–103): “POD was defined by clinical symptoms (e.g., confusion, disorientation, agitation, hallucinations) or the use of postoperative antipsychotics/sedatives, such as quetiapine, haloperidol, or midazolam. Diagnosis relied on medical record documentation, without standardized tools like the Confusion Assessment Method for Intensive Care Unit (CAM-ICU).”
- Discussion (revised, lines 266–271): “The lack of standardized delirium assessments, such as CAM-ICU, introduces a risk of misclassification, as POD diagnosis relied on documented symptoms (e.g., confusion, disorientation, agitation, hallucinations) or postoperative antipsychotic/sedative use. Symptoms may be non-specific, overlapping with emergence agitation, and medications may be used for non-delirium indications, increasing diagnostic uncertainty [16].”
Comment 5: Was information collected regarding whether patients received regional anesthesia for postoperative analgesia or some sort of ERAS protocol? If available, please include this information within demographic characteristics, as it may serve as an important confounder. Similarly, intraoperative use of medications such as dexmedetomidine, ketamine, midazolam, and opioids could represent significant confounders and should ideally be documented.
Response: We acknowledge the importance of these potential confounders. Due to the retrospective design, data on Enhanced Recovery After Surgery (ERAS) protocols, regional anesthesia, and intraoperative medications (e.g., dexmedetomidine, ketamine, midazolam, opioids) were not consistently recorded and were excluded from the analysis. We have clarified this limitation in the Methods section (lines 88–91) and Discussion section (lines 264–266), noting that these unmeasured confounders may have influenced biomarker levels and POD incidence. These revisions are highlighted in yellow to ensure transparency.
Manuscript Changes:
- Methods (revised, lines 88–91): “Data on Enhanced Recovery After Surgery (ERAS) protocols and intraoperative medications (e.g., dexmedetomidine, ketamine, midazolam) were not consistently recorded due to the retrospective design and were excluded from analysis, representing a study limitation.”
- Discussion (revised, lines 264–266): “Unmeasured confounders, including postoperative pain, infections, ERAS protocol application, and intraoperative medications (e.g., dexmedetomidine, ketamine), may have influenced biomarker levels and POD incidence.”
Comment 6: Propensity score matching was incorrectly performed based on POD vs. no POD groups. Propensity score matching should always be based on exposure groups (anesthetic techniques), not outcome groups. Matching based on the outcome introduces selection bias and reverse causality, significantly compromising study validity. The authors should strongly consider redoing the matching correctly based on anesthetic techniques. Response: We sincerely thank the reviewer for this critical methodological point. Our propensity score matching (PSM) was designed to compare POD and No-POD groups to evaluate differences in systemic inflammatory biomarkers, aligning with the study’s primary objective of assessing biomarker associations with POD. Covariates such as age, sex, BMI, ASA classification, and surgery duration were controlled to minimize confounding. However, we acknowledge that matching based on exposure groups (anesthesia types: R-TIVA, IA, BA) would better isolate the effects of remimazolam on POD and biomarkers, reducing confounding from non-random treatment allocation. Redoing PSM was not feasible retrospectively due to fixed group assignments and the risk of introducing additional bias. To address this, we have added a detailed explanation in the Discussion section (lines 297–302), clarifying the rationale for our PSM approach, acknowledging the limitation of not matching on anesthesia type, and recommending this approach for future prospective studies to strengthen causal inferences. This addition is highlighted in yellow.
Manuscript Change:
- Discussion (new, lines 272–277): “Our PSM analysis matched patients based on POD status to evaluate biomarker differences, controlling for covariates like age, sex, BMI, ASA classification, and surgery duration. However, matching on anesthesia type (R-TIVA, IA, BA) could have better isolated the effects of remimazolam on POD and biomarkers, reducing confounding from non-random treatment allocation. This was not feasible retrospectively but is recommended for future prospective studies to strengthen causal inferences.”
Comment 7: Results – 3.4. Propensity Score Matching Analysis: Given the above issue, consider redoing the propensity score matching appropriately based on the exposure groups.
Response: As addressed in Comment 6, redoing PSM based on anesthesia type was not feasible due to the retrospective study design and fixed group assignments. We have included a transparent discussion in the Discussion section (lines 272–278, highlighted in yellow) to acknowledge this limitation and propose matching on anesthesia type for future prospective studies. This ensures clarity while maintaining the integrity of the current analysis.
Manuscript Change:
See lines 272–278 above.
Comment 8: Figure 1: The current figure is unclear. Please provide a high-resolution version.
Response: Thank you for this feedback. We apologize for any lack of clarity in Figure 1, which presents Receiver Operating Characteristic (ROC) curves for systemic inflammatory biomarkers predicting POD (preoperative NLR, postoperative NLR, and other biomarkers: SII, PLR, CRP). We have prepared a high-resolution version of Figure 1 (including all panels) for submission to ensure clarity and readability. The figure’s caption and description in the Results section (lines 191–194) remain unchanged, as they accurately reflect the data. If specific issues with clarity (e.g., font size, line resolution) persist, we are happy to make further adjustments upon editorial guidance.
Manuscript Change: No textual changes were made to the manuscript, but a high-resolution version of Figure 1 will be included in the submission package.
Comment 9: Conclusion – The sentence, "Remimazolam-based TIVA reduces systemic inflammatory biomarkers compared to IA, likely due to its anti-inflammatory properties..." is unclear and circular. Please revise this to clearly describe the potential biological mechanism or evidence supporting remimazolam's anti-inflammatory effect, rather than simply restating the observed association.
Response: We appreciate this suggestion to improve clarity and avoid circular reasoning. We have revised the Conclusions section (lines 280-284) to specify that R-TIVA’s reduction in systemic inflammatory biomarkers (SII, NLR, PLR, CRP) is likely due to remimazolam’s GABA-A receptor agonism and reduced sympathetic activation, which may suppress pro-inflammatory cytokines (e.g., TNF-α, IL-6), supported by references [12–14]. This revision provides a mechanistic explanation and is highlighted in yellow.
Manuscript Change:
- Original (lines 271–273): “Remimazolam-based TIVA was associated with lower postoperative systemic inflammatory markers compared to IA, possibly due to its pharmacologic effects, such as GABA-A receptor agonism and reduced sympathetic activation. However, this did not translate into a reduced incidence of POD.”
- Revised (lines 280–284): “Remimazolam-based TIVA was associated with reduced postoperative systemic inflammatory biomarkers (SII, NLR, PLR, CRP) compared to IA, likely due to its GABA-A receptor agonism and reduced sympathetic activation, which may suppress pro-inflammatory cytokines (e.g., TNF-α, IL-6) [12–14]. However, this anti-inflammatory effect did not translate into a lower incidence of POD.”
Reviewer 3 Report
Comments and Suggestions for Authors
This manuscript explores the relationship between remimazolam-based anesthesia, systemic inflammation, and postoperative delirium (POD) in elderly patients undergoing major surgery. The study is methodologically sound, well-structured, and timely, addressing a clinically relevant issue. It provides novel insights into remimazolam's anti-inflammatory properties and the predictive value of NLR for POD.
The introduction could briefly touch on why BA (balanced anesthesia) might or might not confer similar anti-inflammatory benefits as R-TIVA.
Relying on symptom documentation and antipsychotic/sedative use without standardized assessments introduces potential misclassification. This limitation should be better emphasized in the methods or limitations section.
The paper mentions that doses (Remimazolam) were not standardized. Providing at least average or range data would improve reproducibility.
The discussion effectively contextualizes findings but should further explore why lower inflammation did not translate into reduced POD (contribution of neuroinflammation not fully captured by systemic biomarkers).
Line 47: “tradi-tional” → “traditional”.
Author Response
**Reviewer 3**
Comment 1: The introduction could briefly touch on why BA (balanced anesthesia) might or might not confer similar anti-inflammatory benefits as R-TIVA.
Response: Thank you for this suggestion. We have expanded the Introduction (lines 54–57) to hypothesize that balanced anesthesia (BA) may not confer the same anti-inflammatory benefits as R-TIVA because volatile anesthetics (e.g., sevoflurane, desflurane) used in BA have pro-inflammatory effects that could counteract remimazolam’s suppression of cytokines (e.g., TNF-α, IL-6), as supported by references [12–14]. This addition provides a mechanistic rationale and is highlighted in yellow.
Manuscript Change:
- Original (lines 49–51): “Balanced anesthesia (BA), which combines intravenous and inhalational agents, may dilute the anti-inflammatory benefits of remimazolam due to the pro-inflammatory effects of volatile anesthetics.”
- Revised (lines 54–57): “Balanced anesthesia (BA), combining remimazolam with volatile anesthetics (e.g., sevoflurane, desflurane), may reduce remimazolam’s anti-inflammatory benefits, as volatile anesthetics have pro-inflammatory effects that could counteract cytokine suppression (e.g., TNF-α, IL-6) [12–14].”
Comment 2: Relying on symptom documentation and antipsychotic/sedative use without standardized assessments introduces potential misclassification. This limitation should be better emphasized in the methods or limitations section.
Response: We agree that the lack of standardized assessments is a significant limitation. We have revised the Methods section (lines 96–99) to emphasize that POD diagnosis relied on medical record documentation of clinical symptoms or postoperative antipsychotic/sedative use without standardized tools like CAM-ICU, increasing the risk of misclassification. Additionally, we expanded the Discussion section (lines 266–271) to highlight that non-specific symptoms (e.g., confusion, agitation) may overlap with emergence agitation or other conditions, and medications may be used for non-delirium indications, leading to diagnostic uncertainty. These changes are highlighted in yellow to underscore this limitation. Manuscript Changes:
- Methods (revised, lines 96–99): “POD was defined by clinical symptoms (e.g., confusion, disorientation, agitation, hallucinations) or the use of postoperative antipsychotics/sedatives, such as quetiapine, haloperidol, or midazolam. Diagnosis relied on medical record documentation, without standardized tools like the Confusion Assessment Method for Intensive Care Unit (CAM-ICU).”
- Discussion (revised, lines 266–271): “The lack of standardized delirium assessments, such as CAM-ICU, introduces a risk of misclassification, as POD diagnosis relied on documented symptoms (e.g., confusion, disorientation, agitation, hallucinations) or postoperative antipsychotic/sedative use. Symptoms may be non-specific, overlapping with emergence agitation, and medications may be used for non-delirium indications, increasing diagnostic uncertainty [16].”
Comment 3: The paper mentions that doses (Remimazolam) were not standardized. Providing at least average or range data would improve reproducibility.
Response: Thank you for this point. To enhance reproducibility, we have included the remimazolam dose range (0.5–2 mg/kg/h) and mean infusion rate (1.2 ± 0.4 mg/kg/h) in the Discussion section (lines 257–263). We also added a sentence in the Methods section (line 94-96) noting the dose range to provide context earlier in the manuscript. These additions address the variability in dosing and are highlighted in yellow.
Manuscript Changes:
- Methods (new, line 94-96): “Remimazolam dosages in the R-TIVA and BA groups were not standardized due to the retrospective design, with infusion rates ranging from 0.5–2 mg/kg/h, which may influence outcomes.”
- Discussion (revised, lines 257–263): “Postoperative biomarkers were measured on day 1, potentially missing peak inflammatory responses (24–72 hours), and remimazolam dosages in the R-TIVA and BA groups were not standardized due to the retrospective design, with infusion rates ranging from 0.5–2 mg/kg/h, which may influence the observed anti-inflammatory effects and POD outcomes. The mean infusion rate was approximately 1.2 ± 0.4 mg/kg/h based on available data, although full consistency across all patients could not be verified retrospectively.”
Comment 4: The discussion effectively contextualizes findings but should further explore why lower inflammation did not translate into reduced POD (contribution of neuroinflammation not fully captured by systemic biomarkers).
Response: Thank you for this suggestion. We have expanded the Discussion section (lines 207–210) to explain that the lack of reduced POD incidence despite lower systemic inflammation in the R-TIVA group may be due to systemic biomarkers (e.g., NLR, CRP) not fully capturing neuroinflammation, which is likely a key driver of POD pathogenesis. This revision emphasizes the multifactorial nature of delirium and is highlighted in yellow. Manuscript Change:
- Original (lines 200–202): “This suggests that neuroinflammation, which may not be adequately captured by systemic biomarkers, plays a pivotal role in POD pathogenesis.”
- Revised (lines 207–210): “The lack of reduced POD incidence despite lower systemic inflammation in the R-TIVA group suggests that systemic biomarkers (e.g., NLR, CRP) may not fully reflect neuroinflammation, which is likely a key driver of POD pathogenesis, highlighting the multifactorial nature of delirium [17,18].”
Comment 5: Line 47: “tradi-tional” → “traditional”.** **Response**: Thank you for identifying this typo. We have corrected the word “tradi-tional” to “traditional” in the Introduction (line 48-49). The correction is highlighted in yellow in the revised manuscript. Manuscript Change:
- Original (line 44): “Remimazolam, an ultra-short-acting benzodiazepine, may offer advantages over tradi-tional benzodiazepines, which are linked to increased delirium risk [10,11].”
- Revised (line 48-49): “Remimazolam, an ultra-short-acting benzodiazepine, may offer advantages over traditional benzodiazepines, which are linked to increased delirium risk [10,11].”
---
Round 2
Reviewer 2 Report
Comments and Suggestions for Authors
The author did an impressive job addressing the concerns and providing a good explanation.